# A Study on the Relationship between Mental Well-Being and Cultural Tourism Guides Based on the Interview Methodology

**DOI:** 10.3390/ijerph182413054

**Published:** 2021-12-10

**Authors:** Junsoo Kang, Youngmin Song

**Affiliations:** Department of Tourism, College of Social Sciences, Anyang University, Incheon 23038, Korea; kangminsk@anyang.ac.kr

**Keywords:** mental well-being, cultural assets, cultural tourism guide, storytelling technique, emotional stability

## Abstract

The purpose of this study was to find a way for modern tourists to enjoy increased well-being while being provided with high-quality information about cultural assets. In order for tourists to enjoy well-being, cultural tourism guides must provide quality services while using various storytelling techniques. As the number of tourists who are interested in cultural assets and use their leisure time for this purpose increases, quality cultural tourism commentary can be directly connected to the well-being of tourists. Modern tourists can experience richness of life and emotional stability while being provided with cultural tourism commentary services through various storytelling techniques rather than professional knowledge. In order for tourists to effectively experience well-being through cultural tourism commentary, cultural tourism guides need to implement the following effective commentary. First, culture tourism guides should try to have sense of unity with visitors. Second, they should encourage humanistic imagination through related information. Third, they should provide customized explanations for tourists’ understanding because tourists consist of various classes and ages. Cultural tourism guides who attract tourists’ interest, have appropriate wit, skillful responses to cope with unexpected situations, cheerful laughter, and a loud voice gained satisfaction from many tourists. In modern society, cultural tourism commentators are not limited to simply explaining tourist destinations, but play an important role in satisfying the well-being of modern tourists who seek leisure and emotional stability. The external environment that refers to outdoor atmospherics is also crucial as it influences visitor experiences. In museums, external physical environment factors such as architectural style, positioning of entrances, and exterior décor and signage can be a crucial facet of external ambiance affecting visitor experiences. The external environment (e.g., spacious design, pretty landscape design, outdoor natural surroundings) is a constituent of the tangibility aspect of museum performances. These external environment factors and internal factors together create perceived physical environments that visitors and staff cognitively/emotionally/physiologically respond to. The significance of this study is that various cultural storytelling activities performed by tour guides are examining the possibility of experiencing increased psychological well-being while making tourists as well as themselves aware of the happiness and joy of life.

## 1. Introduction

Modern tourists who travel to various tourist attractions are interested in mental well-being through leisure and emotional stability while being provided with interesting and effective cultural tourism commentary services. Modern tourists with diverse personalities, i.e., cultural tourism consumers, need various types of commentary services that cultural tourism guides can provide. Tourists can get away from their daily life and gain a sense of life satisfaction along with various daily pleasures through the culture and history of a tourist destination. Therefore, cultural tourism commentary can be said to be an important medium through which modern tourists can experience positive mental well-being.

Modern tourists, who value personal leisure, special experiences, and psychological stability, are not interested in a guide’s explanation using the simple knowledge transfer method in the process of cultural tourism. Pearce compares the tourist experience with an orchestra performance in his analysis of tourist behavior [1], meaning that various elements must be combined and harmonized to satisfy tourists. In this regard, cultural tourism guides should use various interpretive techniques such as storytelling to provide effective tourism services to tourists. The storytelling technique is an effective technique that can stimulate the interest of tourists [2].

Thus, various commentary techniques using storytelling are key means to satisfy the cultural tourism needs of tourists. These results have already been verified in several studies [2,3,4]. The discussion of the existing cultural tourism commentary relates to the function and role of the interpretation service [5,6,7] and the cultural tourism commentary [8,9,10,11,12,13] for specific regions. The distinctiveness of this study is that it focuses on the connection between cultural tourism and tourists’ perception of mental well-being rather than the interpretive function of the cultural tourism guide.

Looking at the preceding studies, it can be seen that they have been conducted based on the satisfaction of the interpretive service and the commentary on cultural assets located in a specific region. The implication of this study is that unlike previous research, it will closely discuss the technical aspects of effective and practical commentary of cultural tourism guides. The technical aspect of this commentary is important for cultural tourism guides because effective commentary techniques can serve to help tourists better understand the value and meaning of culture and tourism resources.

In particular, humor can be a great way to maximize the experience of tourists in tourist destinations [14]. This study was conducted mainly using cultural tourism guides who perform cultural tourism commentary for tourists in the Hanyangdoseong Program. Hanyangdoseong, Seoul City Wall was a fortress built during the Joseon Dynasty (1392–1910) to mark the boundaries of the capital and to defend against external enemies.

Thus, cultural tourism guides need to use various commentary techniques such as storytelling in order to provide effective tourism services to tourists.

## 2. Conceptual Framework

### 2.1. Cultural Commentary and Previous Research Analysis

Mental well-being refers to an individual’s perception of life and a state of satisfaction in psychological aspects [15]. Modern people tend to pursue mental well-being, as life and quality of life are improved. Mental well-being is a concept related to life satisfaction and positivity [16]. Modern people need to have the opportunity to experience joy in their daily life in order to experience mental well-being [17].

Awareness of personal mental well-being comes from positivity in life [18]. Therefore, this study aims to understand the possibility that modern tourists can experience mental well-being through cultural tourism. The cultural tourism guide of a modern concept provides tourists with fun and enjoyment rather than simply imparting knowledge [19].

Cultural tourism can draw the attention of tourists according to the method of interpreting various cultural heritages [9]. Reference [20] mentions the communication ability and situational coping ability of cultural tourism guides as factors that can attract tourists’ attention. Essentially, in cultural tourism, commentary has a great influence on the satisfaction of tourists [3]. [4] discusses tourist satisfaction according to the cultural tourism commentary service method.

Tilden pays attention to the following about cultural tourism: First, the need for a representative delivery method to increase the understanding of tourists is raised [21]. Second, tourists have diverse personal preferences in cultural tourism [22]. Cultural tourism needs to devise various techniques to satisfy the needs of tourists with various personalities [23].

These various commentary techniques can provide many possibilities for the cultural tourism industry [24]. Cultural tourism can improve the quality of life of tourists [25]. In other words, cultural tourism can be a means to increase tourists’ satisfaction with life [26]. This means that the role of cultural tourism guides can play an important role in tourist satisfaction [27].

Cultural tourism guides need to use storytelling techniques as a means to satisfy tourists. The techniques of storytelling include modified and processed storytelling, creative storytelling, real storytelling, and storytelling in the form of music [28]. Modified and fabricated storytelling includes parody storytelling and rumor storytelling [29]. The former is a technique of humorously expressing the material of a specific work by mimicking an author’s style, while the latter is a type of advertisement in which a company or brand creates stories that consumers may be interested in.

Creative storytelling includes series storytelling, dream storytelling, and imaginary storytelling [30]. Series storytelling is to develop a series of delivery contents using a subject or product as a material. Imaginary storytelling is a technique that utilizes an imaginary world to use unrealistic stories as advertisements [31]. Real storytelling includes episodic storytelling and experiential storytelling [32].

The former is a technique to approach consumers in a friendly way using everyday stories as the subject, and the latter is a technique to increase product reliability by delivering advertisements that consumers have directly experienced back to consumers. Storytelling in the form of music includes storytelling in the form of songs and storytelling in the form of music dramas [33]. The former is not just an advertisement with background music, but a product message that is included in the song in the advertisement. The latter is storytelling in the form of advertisements in which music, drama, and products harmonize in the form of a music video.

A cultural tourism guide is an expert who understands the tastes of tourists with various psychological and personalities and provides commentary services through appropriate expressions such as verbal actions and gestures [34]. Therefore, the directions that cultural tourism guides should pursue should be able to provide tourists with the enjoyment and leisure [35]. In order to effectively conduct these discussions, this study recorded in-depth interviews with cultural tourism guides on the spot in writing and suggested common requirements based on them.

### 2.2. Efforts to Satisfy and Interest Tourists

Cultural tourism guides need to have various commentary techniques as well as specialized knowledge on the subject of commentary. In one study [2], it was recognized that a cultural tourism guide’s ability to deliver commentary is related to the satisfaction of tourists. According to [36], the commentary method of cultural tourism guides can give tourists a negative or positive perception.

According to the study of [37], the curiosity and satisfaction of tourists comes from the vivid and interesting commentary of cultural tourism guides. Storytelling is a technique that cultural tourism guides can use effectively. Storytelling allows an individual to experience an emotional feeling called catharsis [38]. This is because the storytelling technique gives the story a sense of realism and liveliness, while allowing people to experience fun and enjoyment [39]. Therefore, cultural tourism guides need to use various storytelling techniques in order to perform effective commentary.

This study identified common factors through in-depth interviews and participatory observations with cultural tourism guides, who gave tourists a sense of pleasure and happiness. First, cultural tourism guides, who were popular with tourists, drew attention and pleasure from tourists through witty words, gestures, positive reactions, and funny expressions in common. The methods of the cultural tourism guide increased the satisfaction with cultural tourism by forming intimacy and sympathy with tourists. The satisfaction of these tourists can be a factor in raising awareness of mental well-being as it leads to a positive attitude toward life. The satisfaction of tourists affects the competitiveness of cultural tourism guides [40]. Cultural tourism guides need to satisfy the needs of tourists through differentiation strategies [41]. One of the strategies for securing competitiveness of cultural tourism guides is the use of storytelling [42]. Cultural tourism guides must have the ability to transform historical knowledge into stories that can resonate with tourists [43]. Modern tourists can experience emotional stability and life satisfaction through cultural tourism [44]. Therefore, cultural tourism guides need to secure various narrative techniques that can satisfy the needs of tourists [45]. This is because satisfaction with cultural tourism can be directly linked to the mental well-being of tourists.

The efforts of cultural tourism guides do not show consistent results [26]. This means that cultural tourism guides need the ability to quickly identify the needs of tourists [46]. This is because cultural tourism guides do not simply transfer knowledge, but promote the mental well-being of tourists [47].

Cultural tourism guides need to provide not only commentary but also a friendly attitude and various services to tourists. The kindness of cultural tour guides and the provision of various services can raise the expectations for mental well-being while promoting emotional stability of tourists. This is because cultural tourism commentary is an activity that allows tourists to enjoy their daily lives and acquire emotional stability based on the meaning and value of tourism resources [48]. In order to increase satisfaction, cultural tourism guides need to perform comfortable and natural explanations for the psychological stability of tourists [49]. Tourists’ positive image and satisfaction with tourism are closely related to the role of cultural tourism guides [50].

## 3. Methods

### 3.1. Research Subject

This study discusses the life satisfaction and mental well-being that tourists can experience through cultural tourism. Basically, in order for tourists to recognize mental well-being through cultural tourism, the role of cultural tourism guides is important. Therefore, this study will conduct a discussion focusing on the correlation between the interpretation technique of cultural tourism guides and the satisfaction of tourists.

In this study, through in-depth interviews with cultural tourism guides, the contents of the work of cultural tourism guides and how to perform them effectively were investigated. From 1 September to 31 October 2019, in-depth interviews were conducted with cultural tourism guides. Since the in-depth interviews were conducted before the outbreak of COVID-19, the researchers were able to hear the various stories of cultural tourism guides in depth.

In-depth interviews in this study were conducted individually for 2–3 h. The researchers also used phone calls or e-mails to ask additional questions when needed. The structure of the questions were: (1) Commentary method; (2) Tourist response; (3) Tourist satisfaction; (4) Effective cultural heritage commentary strategy; (5) Culture tourism and mental well-being.

The Hanyangdoseong program is hosted by the Jung-gu Office in Seoul and has about 40 cultural tourism guides in four sections centered in the old Hanyangdoseong. Hanyangdoseong, Seoul, has four sections: Baekak section (Changuimun to Hyehwamun), Naksan and Heunginjimun (Hyehwamun to Jangchung Gymnasium), Namsang (Mokmyeok Mountain) and Sungnyemun (Jangchung Gymnasium to Donuimun Site), and Inwang Mountain section (Donuimun Site to Changuimun) [45].

In 596 (1396), Hanyangdoseong was renovated several times after being built along the ridges of Baekak (Bukak Mountain), Nakta (Nak Mountain), Mokmyeok (Nam Mountain), and Inwang’s Naesa Mountain. Hanyangdoseong, with an average height of about 5–8 m and a total length of about 18.6 km, served as the longest (1396–1910 years) fortress in existence [51].

These researchers conducted in-depth interviews with 11 cultural tourism guides who work in Hanyangdoseong. In addition to in-depth interviews with cultural tourism guides, these researchers also conducted close participation observations. This is because in-depth interviews are one of the important means to achieve the purpose of research [52]. The cultural tourism guides, i.e., the subjects of the in-depth interviews, were mainly in their 50s and 60s, and they were evenly distributed with 6 males and 5 females. In addition, the experience of cultural tourism guides was relatively evenly distributed between 3 and 14 years. Two researchers not only conducted in-depth interviews with cultural tourism guides, but also checked the responses of tourists closely for two months.

Cultural tourism guides work in various tourist areas, and each commentary space is categorized as a course. In Table 1, the commentary course means a space for each culture tour guide to explain. As can be seen from Table 1, the interviewees who were cultural tourism guides were elderly people who were freed from their daily lives by retiring or raising children. Therefore, these cultural tourism guides were doing business with the intention of living healthily with the spirit of serving for the rest of their lives rather than making money.

As mentioned earlier, the in-depth interviews for this study were held in 2019, and considering that it takes a long time, the researchers conducted the in-depth interviews from June 25 to 29, 2021 while strictly observing the COVID-19 safety rules. The in-depth interviewers were not replaced, and there were no major difficulties in conducting the in-depth interviews as they continued to work with cultural tourism guides.

In the interview, the order of questions was adjusted according to the response flow of the interviewees, and results were derived based on the results of collecting opinions from the target group through in-person interviews and phone calls. When writing the improvement plan, we gathered common and unified opinions, and after that, interviews were conducted over the phone again or reviewed to supplement the interview contents if necessary.

This study resulted in common and unified opinions and we also attempted to make phone calls or revisit participants in order to supplement the interviews. The procedure was conducted with the consent of the interviewees to ensure the recording of the interview and anonymity. Since the content of the in-depth interview was mainly related to the commentary technique and was not significantly different from 2019, duplicate content was not included in this study, and additional discussions were presented as follows. The main contents of the questions were about mental well-being and cultural tourism.

Additionally, this study focused on mental well-being and conducted in-depth interviews with cultural tourism guides. In addition, this study briefly recorded all the opinions of tourists who participated in cultural tourism at the time regarding spiritual well-being. Based on those results, this study tried to analyze the opinions and thoughts of tourism experts (two professors, two tourism industry officials), comprehensive cultural tourism guides, and tourists from the viewpoint of mental well-being.

### 3.2. Analytical Process of Research

Cultural tourism guides perform various functions to satisfy the needs of tourists who want to enjoy leisure and find emotional stability. This means that the role of cultural tourism guides is not limited to simply explaining cultural resources. This study focused on the role of cultural tourism guides leading cultural tourism in order to satisfy the needs of modern tourists who value leisure and psychological stability.

This study also focused on the possibility that tourists’ satisfaction with cultural tourism could be connected to their perception of mental well-being. Therefore, this study conducted a discussion on various commentary abilities of cultural tourism guides that can satisfy the needs of tourists. Based on these discussions, this study tried to draw results by closely examining the responses of tourists through in-depth interviews with cultural tourism guides and participatory observations.

## 4. Tourists’ Mental Well-Being Experiences through Effective Commentary

### 4.1. The Wit of the Guide and the Pleasure of Tourists

Actor D started the commentary in a strong tone, but many tourists were distracted and were talking to each other. However, Actor D did not care and said:

Actor D: “This is a tree that has lived for over 1000 years. The tree’s name is “Seonbi Tree”. By the way, what does Seonbi mean in English? Who, who? Tourist: “ Scholar”.

Actor D: Uh, yes. Yes! It is a scholar tree (Seonbi in Korean is scholar in English)”!

Actor H: “Hanyangdoseong is a total of 18 km long, reaching the Bukak mountain to the north, the Nak mountain to the east, the Inwang mountain to the west, and the Nam mountain to the south. In 1392, Lee Seong-gye (the great king of Joseon) was Jeonju Lee, but do any of the people here have Jeonju Lee’s surname? Oh, a lot of Jeonju Lees came today. These are all cultural properties of your immediate ancestors. Tourist: “Really. I am proud of my ancestors. Please explain well”.

Actor I: “During the Joseon Dynasty, there was a saying that a person who participated in the government-administered exam would pass the exam if they went around Hanyangdoseong. If any of the people here are taking important exams, go around this place”.

Tourist: “Wow! My son has to go to university next year, and he has to go around Hanyangdoseong”!

Actor I: “Then, your son will definitely pass the university exams”! (laugh)

(All the tourists are smiling, having fun and being happy.)

Actor J: Despite being a strict society, King Jeongjo, the 22nd king of the Joseon Dynasty, granted the children of concubines of all aristocrats the qualifications to take the test to obtain office. Therefore, people who came to Hanyangdoseong after taking past exams in each region with a dream of obtaining an official position at the time believed that they would pass around Hanyangdoseong village once. There are many parents and students today, so everyone who comes here should have great enthusiasm for studying at Hanyangdoseong. (laugh). Tourist: During the Joseon Dynasty, which had a strict system of status, King Jeongjo was an awakening king. As a parent, I am very happy to learn about our educational history through this time with our children.

Actor K: Inwang mountain is a place to tell the story of a tiger and his devoted son, Park Tae-sung. Park Tae-sung had to cross the Mooak hill of Inwang mountain every day to find his father’s grave at the foot of Samgaksan Mountain. Then one day, a tiger, impressed by Park Tae-sung’s filial piety, saved him so that he could safely go to his father’s graveyard without eating him. The beast, the tiger, sincerely believed in Park Tae-sung’s words that he was the child of his deceased father, and had bowed with him in the tomb. The tiger thought that Park Tae-sung was his brother. So, when Park Tae-sung died, it is said that the tiger died together by the tomb. Your parents who have come here will soon die, and you will also die someday. When your parents are alive, do a lot of filial piety. Anyway, we all die one day. (Some tourists cry): This is a moving story. It was the first time I knew that this kind of story was handed down to Inwang Mountain. Yes. We all die someday, but I think we have to live well while we live.

As a result of the participation observation, Actor D approached the tourist who answered the feature he presented, held his hand, and smiled like a child, showing excitement. Other tourists who were watching this scene showed interest in Actor D’s commentary. All tourists loved it, applauding and cheering. It was a moment when guides and tourists felt a sense of unity. On the same day, Actor H brought out the surname of the tourist and the surname of the king of the Joseon Dynasty as a common interest. Among the many surnames in Korea, the distribution of Lee surnames is widespread, so Actor H succeeded in empathizing with tourists based on the full support of tourists as a common surname. Figure 1 shows a tour guide explaining cultural relics to tourists in Hanyangdoseong.

Actor I used the stories handed down in Hanyangdoseong to deliver a positive message so that tourists could feel the comfort, change of mood, and happiness of reality. Such wit of Actor I can increase satisfaction with life in that it makes tourists feel good, regardless of the credibility of the story. These tourists’ life satisfaction can be related to their mental well-being.

Actor J found out that the tourists who participated that day were mainly parents and students, and focused on the government office examination during the Joseon Dynasty and succeeded in attracting the attention of parents and students who are interested in studying. Actor K created a consensus by stimulating the common filial piety of mankind while focusing on the real problems of middle-aged tourists. The actors D, H, J, and K each had their own personality, but the common point on which they could draw attention from tourists was the ability to actively communicate and empathize with tourists.

In the end, the approach to tourists differed for each cultural tourism guide, but the common feature of guides who were popular among tourists is the ability to consistently gain a sense of unity with tourists. Actors D, H, J, and K were able to maintain constant concentration after drawing out common interests among tourists. Tourism satisfaction refers to perceived satisfaction in tourism experience [53]. There is a correlation between preferences for tourist attractions and intention to visit [54].

Even when the same cultural property is described, the degree of interest of tourists varies depending on how a cultural tourism guide describes it. For example, differences were also revealed in the section describing how to build walls. The construction method of the wall depends largely on whether it was built during the three periods of King Taejo, Sejong and King Sukjong. The walls of King Taejo’s era had wide stones on the bottom and small stones on the top. The walls of King Sejong had rectangular stones on the floor, and debris was mixed between them. King Sukjong’s walls were made of ‘ㄱ’ and ‘ㄴ’ letters rather than perfect squares to increase friction between stones to reinforce the wall [55]. The following is a comparison of Actor A and Actor C describing the structure of the wall. Actor A was too faithful in their explanation:

Actor A: “The wall of King Taejo was built by roughly processing natural stone. King Sejong’s wall was made of many stone materials, unlike King Taejo. The next wall of King Sukjong was built by standardizing stones close to a 45 cm square”. 

Actor C: “What kind of king is Taejo? When he was a subordinate general of the king of the Goryeo Dynasty before ascending the throne, he violated the king’s order to conquer Wihwa Island and killed the king of Goryeo with his soldiers. As Joseon Dynasty was newly established, there were too many things that were not equipped and there were too many things to do. As a result, these times were applied to the building of walls, and the walls were roughly built up at that time. Everyone knows who Sejong the Great is? He is the king of Korean words and is still respected. The stones on the walls of the city at this time were not roughened. The next king, King Sukjong, was charismatic as a historical friend. For example, he was Danjong. To restore the honor of the king and the sergeants (four servants who died to protect King Danjong), they completed the border with the Qing Dynasty (the last dynasty of China), who frequently occupied Mount Baekdu, and acquired sovereignty over Dokdo Island, a Korean territory. A person, Anyong-bok, was from the era of Sukjong’s reign. King Sukjong did very well in politics. This was also applied to the wall stone, and the wall stone at this time was made in a standard size of about 45 cm square on one side”.

Actor E: “At that time, the paintings generally drawn on the gates were drawn as dragons, white tigers, red birds, and black turtles, indicating the direction of the north, south, east, west, and north. Why? (Laughter) At that time, in the hot summer, did Changuimun people enjoy chicken like modern people like chicken and beer? (Laughter) Tourist: Where is the chicken at that time? It’s a very interesting thought”. (Everyone laughs).

Actor E: “Really? (laughs) The answer can be found in the fact that the village of Hanyang Castle in the south of Changimun is curved like a centipede. It was a wish”. (laugh)

Actor F: “Sukjeongmun, the large gate to the north of Hanyang Castle Village, was also called Sukcheongmun. It means the narrowing of the northern border to make the city comfortable. At Sukjeongmun, a ceremony was held to clear the sky after the rainy season. Inside, there is Samcheong-dong (Samcheong-dong means a village that has three kinds of cleanliness.) This village became Samcheong-dong because it has clear mountains, clear water, and the clear spirit of the people living there”.

According to the observations of the participants, while listening to Actor A’s explanation, tourists became bored and began to make noise. Tourists did not listen to Actor A’s explanation and instead took pictures or chatted with each other. Tourists seemed to need more than just knowledge of the walls. Meanwhile, Actor C was good enough to capture the attention of tourists while explaining the same wall. Actor C greeted tourists with a smile and humor before explaining the walls, and tried to get to know them as much as possible.

Actor C tried to provide a vivid story, not just a commentary, while explaining the walls. This storytelling technique allowed tourists to learn a wealth of stories beyond their prior knowledge of the walls and to remember the guide’s expertise for a long time. According to participant observations, while listening to Actor C’s story, tourists gave support for his commentary and encouraged it with applause.

Actor E, mixed with outright laughter and good speech, fascinated tourists who were not feeling well. He used the technique of turning hostile elements into enjoyment through cute smiles and gentle actions. Actor F used various tones and high pitches in their descriptions to minimize the boredom of tourists.

In addition, he prepared and linked various stories related to the depicted cultural properties, providing various pleasures to tourists. Cultural tourism guides who can capture tourists through these methods have in common that they plan and construct scenarios themselves, and applied methods to communicate with tourists through stories rather than simply commenting. The efforts of these cultural tourism guides can influence the formation of a local image. The image of the region begins by evaluating the existing image and trying to change it in a new direction [56]. The emotional image of the tourist destination is related to the emotion of the individual, which can increase emotional recollection of past experiences [57].

### 4.2. Communication and Empathy

Culture tourism guides should be able to grasp and understand the value and meaning of culture and tourism resources. This ability of cultural tourism guides not only increases concentration in specific classes, but also means customized commentary skills for tourists of various levels. For example, when referring to employees of a company, it is a method to elicit positive responses through continuous praise and encouragement, considering the high work stress of the employee.

Cultural tourism guides should have the ability to empathize with the tourist level in their commentary technique. Empathy is the ability to accurately predict what another person perceives from the other person’s point of view [58]. In other words, empathy is understanding the other person’s situation and sharing the other person’s feelings and point of view. Cultural tourism guides must have the ability to understand and react emotionally to other people’s experiences in order to give effective commentary to tourists [59].

Actors A, B: “Tourists with children on weekends do not need a simple history commentary. This is because historical knowledge can be fully grasped through schools and internet. We explain history to children. I think it’s not just about doing things, but adding stories to make kids interested in history”.

Actor C: “We have a variety of professions and we didn’t major in real history, so we can’t teach tourists professional history like a historian. We don’t give lectures, we give commentary. Historical contents are those taught in the classroom by professional historians”.

Actor B: “We want to be educated on the skills and attitudes to give effective commentary. History professors who conduct refresher education tend to focus only on theoretical content because they do not know the actual situation in the field. If professors actually explain cultural properties to kindergarteners who have visited the field, will their knowledge be accepted? I think refresher education should be led by people who have actually experienced it in the field. I think I need the practical teaching method that I have”.

Actor G: “When we go to refresher education, history professors focus on high-level expertise. For non-history majors like us, lectures are too difficult and useless. The education we want is a commentary technique with effective storytelling techniques”.

Cultural tourism guides play an important role in empathizing with tourists and conveying emotions in tourist destinations. Culture tourism guides should lead the commentary in an interactive way to communicate emotions with tourists rather than explaining it in just one way. Culture and tourism guides need excellent directing skills to interact with tourists. To carry out this commentary, you need a clear and easy way to convey the topic and core of the commentary.

Cultural tourism guides should be able to lead tourists with tourism needs to recognize the historical and cultural background and value of the cultural heritage and to experience the attraction of the resource. This ability is related to tourism satisfaction. Tourist satisfaction is an emotional result that tourists experience after experiencing a service and subjectively recognizing the result [60].

In the expectation discrepancy theory, satisfaction is concluded based on the difference between an individual’s actual performance and the level of expected performance. It can be said that customer loyalty determines the number of returning tourists [49]. Cultural tourism guides need excellent directing skills to increase the satisfaction of tourists. Tourism satisfaction can be said to be a combination of expectations for tourist destinations and emotional reactions and attitudes through interaction after experiencing the region [61].

The most basic strategy of cultural tourism guides to promote exchanges with tourists in the field is to identify and derive common interests between the two. Culture and tourism guides with excellent outputs draw out common interests and use their own strategies to maintain close relationships with tourists. For example, Actors H, J, and K focused on finding a common base for tourists on site.

After securing the common denominator of tourists, Actors H, J, and K’s directing strategy fused their individuality and maintained a sense of intimacy. Meanwhile, Actor D, from the beginning, used his own strategy to attract tourists’ curiosity and interest. First, Actor D’s voice was unique. Actor D properly used a push and pull technique with tourists. For example, it consistently attracted half of the tourists while also creating persistent problems. Sometimes, Actor D ignored tourists and caught tourists’ attention with excessive praise if their answer was correct. Actor D used open laughter and various questions. In addition, he built a sense of intimacy with the tourists and responded often when tourists answered his questions. In particular, the humor he used in his commentary played an important role in maintaining bonds with tourists. Actor D maintained a sense of unity and intimacy with tourists by repeatedly using these strategies.

When Actor H explained the family tree of the Joseon Dynasty, he drew on a common topic by additionally explaining the relationship between ordinary families. They used a strategy of breaking down the walls of the tourists, drawing consensus among them by adding everyday family stories. Moreover, even though Actor H was a man, he brought out the laughter of tourists with his friendly smile and language skills reminiscent of the image of a friendly neighborhood aunt.

Actor J always tried to maintain a close relationship through a strategy of caring for tourists. At each point of the commentary, the safety and atmosphere of tourists were carefully checked, and the commentary continued in a soft and calm tone. It imprinted the image of motherhood by giving the impression that they cared more about the safety of tourists than the commentary. She further captured the attention of tourists with her meticulous attention to detail. Actor J was praised by tourists for her gentle technique that seemed to embody a fairy tale for young children, rather than being overactive or laughing. Table 2 is the step-by-step mental well-being experience stage for tourists that can occur in the process of providing a tour guide’s commentary service. In Table 2, Stage 1 presents key factors that tour guides can use to experience the mental well-being of tourists. And based on the core services of these tour guides, tourists’ well-being experiences are presented in Steps 1 and 2.

Actor K used a strategy to stimulate the emotions of tourists. In the course of his commentary on history and culture, he presented common interests such as the nature of life and the relationship between parents and children. This technique of Actor K was able to capture tourists’ interest effectively. His key explanatory skill was emotional stimulation.

Various commentary services provided by cultural tourism guides to tourists can provide enjoyment and leisure in life. This leisure and joy of life is the foundation for experiencing positive mental well-being [62]. This is because cultural and tourism guides can experience life satisfaction and self-fulfillment through commentary services, and tourists can experience daily pleasures. Daily small pleasures are closely related to mental well-being [17]. Daily happiness can increase the possibility of experiencing mental well-being while securing life positivity [18].

## 5. Discussion and Implications

According to the various interpretation services provided by cultural tourism guides, there are more opportunities to experience not only satisfaction but also daily leisure and emotional stability. These new daily experiences of tourists raise awareness of mental well-being in life. Therefore, various storytelling techniques and interpretation service skills of cultural tourism guides can have a great impact on the emotional stability and life satisfaction of tourists.

In order to achieve this goal, cultural tourism guides must have a sense of unity, different strategies, and interpretive abilities suitable for the type of tourist. In this study, cultural tourism guides used the various strategies mentioned above to improve not only the concentration of tourists but also their positive attitudes. The implication of this study is that the cultural commentary service of cultural tourism guides is not limited to simply providing information on cultural heritage, but can also have a great impact on the lives of tourists.

Existing prior studies conducted discussions on the form, method, and attitude of cultural tourism guides. On the other hand, this study conducted a detailed and in-depth discussion on the effect of modern tourists, who have increased income levels and increased leisure time to tour various regions in their daily life and receive commentary services. In other words, the implication of this study is that it discussed not only the commentary attitudes of cultural tourism guides, but also the effect on well-being of tourists.

## 6. Limitations

The ability of cultural tourism guides to promote the mental well-being of tourists in this study was found in effective storytelling commentary service and communication skills. Effective explanation means that the subject explaining it and the listener can communicate smoothly and form a consensus [63]. This study conducted a detailed discussion on the role, function, and effect of a cultural tourism interpreter as an important medium that enriches the lives of modern tourists.

The role of the cultural tourism guide is to induce revisiting by drawing a positive image of the region and its cultural properties from tourists [64,65,66]. This study showed that cultural tourism guides’ effective commentary strategies lie in their ability to draw consensus from tourists, to explain rich knowledge about cultural properties for in-depth commentary services, and in their communication skills. This study specifically dealt with various techniques and effective service methods for cultural tourism guides. Nevertheless, the limitation of this study is that it discussed an effective commentary service method for tourists’ mental well-being limited to one specific area.

The cultural tourism commentary service can be applied in various ways and various commentary services can be presented according to the characteristics of the region. However, this study had its limitations as it presented a method of commentary service targeting cultural and tourism guides in one specific region. Future research needs to be conducted while utilizing the interpretation service methods and regional characteristics of cultural and tourism guides in more diverse regions.

## 7. Conclusions

The modern society is an era of complexity, speed, and an abundance of information. In this social atmosphere, tourists want to experience peace of mind, rest, and relaxation. Tourists with diverse personalities want to experience leisure and peace of mind outside of everyday life. The cultural tourism guide should be able to apply the interpretation service while recognizing the various cultural consumption desires of these tourists. Cultural tourism guides not only provide cultural heritage commentary services, but also interact with tourists to acquire a sense of community unity and intimacy. 

This intimacy can be a motif for tourists to experience peace of mind. Because cultural tourism guides do not educate tourists, but provide fun and enjoyment while interacting and interacting with them. Tourists can experience cultural needs and spiritual well-being through various services of the cultural tourism guide. In fact, at the site of the cultural tourism commentary, all the tourists were full of bright expressions and joy. Tourists are experiencing the joy of life by experiencing conversation, empathy, and pleasure with a cultural tourism guide. In other words, in the modern tourism industry, cultural tourism guides perform a function that can provide an experience of mental well-being to tourists with diverse personalities and cultural consumption desires.

## Figures and Tables

**Figure 1 ijerph-18-13054-f001:**
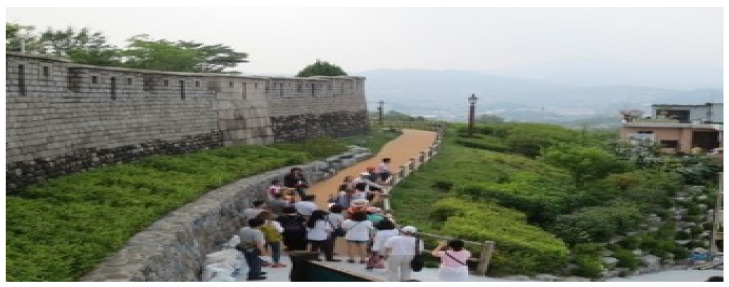
Tourists and cultural tour guide in Hanyangdoseong.

**Table 1 ijerph-18-13054-t001:** Characteristics of the participants.

Order	Actor ^1^	Sex	Age	Commentary Course	Commentary Career
1	A	Male	50s	4 Course	6 years
2	B	Male	50s	1 Course	5 years
3	C	Female	60s	2 Course	9 years
4	D	Male	60s	3 Course	7 years
5	E	Female	60s	1 Course	8 years
6	F	Female	60s	4 Course	13 years
7	G	Female	60s	2 Course	12 years
8	H	Male	50s	3 Course	3 years
9	I	Male	70s	4 Course	14 years
10	J	Female	60s	2 Course	12 years
11	K	Male	50s	3 Course	7 years

^1^—Interviewees, cultural tourism commentators, will now be called Actors.

**Table 2 ijerph-18-13054-t002:** Step-by-step mental well-being awareness process through cultural tourism commentary.

Order	Stage 1 (Guide Service)	Stage 2 (Tourist Experience)	Stage 3 (Tourist Experience)
1	wit	experience of pleasure in everyday life	experience of psychological stability
2	humor	experience of positivity in reality	
3	active reaction	experience of activeness	experience of relaxation
4	interesting story	fresh fun experience	awareness of mental well-being
5	providing intimacy	experience of intimacy	
6	induction of consensus	the experience of building empathy	

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
