# Peer review of "A Study on the Relationship between Mental Well-Being and Cultural Tourism Guides Based on the Interview Methodology"

_ijerph, 2021, doi:10.3390/ijerph182413054_

Round 1

Reviewer 1 Report

I am pleased to have this chance to review this research manuscript for the consideration of publication in this journal. This study aims to investigate how to make tourists enjoy their mental well-being in the context of cultural tourism through tour guides' ability to deliver knowledge and information effectively. The authors employ the qualitative method (i.e., in-depth interview). This study might bring new suggestions to practitioners who are responsible for developing the tour guide's training program.
As a reviewer, however, would like to give the authors several suggestions to improve this manuscript.
1. I would like to suggest that authors should use more appropriate terminologies in the manuscript. For example, who are modern tourists? Is it correct to use "cultural tourism guides"?
2. Could you please provide the specific theoretical backgrounds regarding various storytelling techniques?
3. Please delete the name of the Actors.
4. Please reorganize the contents in your manuscript and remove the redundant contents.
Lastly, I would like to recommend that the authors must proofread this manuscript.

Reviewer 2 Report

This paper analyses how cultural tourism is related with mental well-being using a qualitative study, based on interviews to 11 cultural guides. The idea sounds interesting and relevant for tourism literature, anyway, I will like to show main concerns I have with respect to this manuscript.

# The authors assume that cultural tourism is related to well-being but it could be the other way around, happier people could choose cultural activities. It could also be possible that tourism itself generates well-being, without distinguishing the type of tourism. 

# The main problem of the manuscript is that authors try to analyze customer's well-being using an interview to the tourist guides. There are many biases in this evaluation that invalidate the wellbeing measure. It would be necessary to ask tourists themselves about their own well-being and use appropriate methods. 

# A more thorough review of welfare and its measurement would be necessary.

# I also recommend reviewing how information is presented in qualitative studies.

# What are the reasons for choosing only eleven guides?

# The article has a major problem in the operativization of well-being that generate doubt on all the conclusions reached by the authors.

Reviewer 3 Report

On the surface this seemed like an interesting paper. I liked the idea of linking mental well-being to cultural tourism. However, feel that the paper did not live up to this lofty expectation.

First, there was a lot of repetition throughout the paper. The same ideas were mentioned at numerous points.

Second, the literature review was repetitive and was basically a list of articles that dealt with the subject. In some cases, the authors would list Study A and state what the study was about but not how the findings related to the purpose of this paper.

Third, there are several places where the authors talked about the purpose of the paper, and each time the purpose was slightly different.

Fourth, the methodology section was poorly written. The focus was on tour guide interviews, but then in the next section there was a transcript of different tour guides talking to tourists. The methodology did not match the data that was shown and discussed.

Fifth, I did not see much of a link between tour guiding and mental well-being based on the findings of this paper. If there are links, I did not see them.

Sixth, why were tourists not interviewed for this paper? How can the authors suggest links between tour guiding and mental well-being when tourists were not consulted?

Reviewer 4 Report

Thank you for giving me the opportunity to review the manuscript. Even though the focus of study seems interesting; the manuscript is not yet ready for publication.

The Abstract seems like the recommendation section or part of the literature rather than strongly highlighting the major points of the research and explain why the work is important.

The discussion in the introduction seems very general and dose not strongly discuss what are the research gap/ questions and objectives.

The literature review is very limited. The study implications are very general and hardly revolutionary.  

There are many statements and arguments that are not supported by sources. For example, the introduction starts by stating that “Modern tourists who experience various tourist attractions are interested in mental well-being through leisure and emotional stability while being provided with interesting and effective cultural tourism commentary services. This is because tourists can get away from their daily life and get a sense of life satisfaction along with various daily pleasures through the culture and history of the tourist destination”. It is not clear how and why this happens. The authors refer to modern people/tourists in the manuscript while it is not clear who this people/generations are.

Reviewer 5 Report

The study is very interesting and it highlights the importance of the guide presentation about the touristic objective and the mental well-being of the tourists.

I expected to see a feedback from the tourists and the impact of the guide presentation in a quantitative study.

Round 2

Reviewer 2 Report

Dear authors. In my previous revision we explain main concerns about this paper. Some of them are due to basic aspect, mainly if the measures of variables are or not appropriate.

The authors explain the reasons for the decisions they have made, for example in the number of participants, but the measure of well-being is inappropriate; there is no theoretical or methodological basis for ensuring that the opinion of a third party can measure the well-being of clients. The article is unchanged in the essential criticisms I have made.

Reviewer 3 Report

Please see my comments to the editors.

Reviewer 4 Report

.
